# Importance of Government Roles for Market Expansion of Eco-Village Development Plan Establishment Research: Case Study in the City of Suwon, South Korea

**Soo-Young Moon [1], Daehee Jang [2], Hyeon Soo Kim [1], Ji-Young Lee [1] and Jonghoon Kim [3,\*]**

[1]   Living Environmental Research Center, Korea Institute of Civil Engineering and Building Technology (KICT), Seoul 411-712, Korea; symoon@kict.re.kr (S.-Y.M.); hskim1@kict.re.kr (H.S.K.); raumpl@kict.re.kr (J.-Y.L.)
[2]   Green Building Research Center, Korea Institute of Civil Engineering and Building Technology (KICT), Seoul 411-712, Korea; zzan1113@kict.re.kr
[3]   Deptartment of Construction Management, University of North Florida, Jacksonville, FL 32224, USA
\*   Correspondence: jongkim@unf.edu; Tel.: +1-904-620-2746

**Abstract:** Korean governmental ministries are promoting strategic projects to support cost-saving and low-carbon technologies in residential complexes and commercial buildings in the City of Suwon, South Korea. Suwon City will build throughout the city focusing on economic feasibility by selecting performance targets and using applied technologies for locations to be continuously expanded. This case study of Suwon shows that the local Korean government has prepared a project to spread eco-friendly residential complexes and is trying to introduce and realize eco-friendly construction standards proposed by the central Korean government. The central government is working to actively establish a system to promote eco-friendly construction technologies and encourages people to use eco-friendly construction methods and products. To build the demo-complex in the city, the role of the government was re-examined considering the universalization of energy and environmental technology through the analysis of case studies where these technologies were applied to residential complexes. The objectives of this research study are: (1) to establish a land use plan for the eco-village site in the City of Suwon, (2) to establish an external space plan, which includes the environmental aspects, and (3) to have alternative designs through a multi-criteria decision-making process. This study also used a cost-benefit analysis (BCA) to evaluate and ensure that there was no waste of the Korean government budget contribution, and to observe the business feasibility based on economic performance.

**Keywords:** ecological land use; eco-friendly village; site development; residential

## 1. Introduction

Buildings are the largest consumer of energy in the United States and the largest emitter of greenhouse gas (GHG), accounting for approximately 36% of the entire nation's annual energy consumption [1,2]. The United States accounts for approximately 20% of the world's energy consumption. Buildings consume the same amount of energy consumption as transportation and industry combined [3].

A zero-energy building, also known as a zero net energy (ZNE) building, or a net-zero energy building (NZEB), is a building with zero net energy consumption. The ZNE means the total amount of energy used by the building is roughly equal to the amount of renewable energy created on the site [4,5]. These buildings contribute to less overall greenhouse gases than similar non-ZNE buildings [6].



The aim of these buildings is not only to consume renewable energy and produce fewer greenhouse gases but also to reduce energy consumption and greenhouse gas production elsewhere by the same amount [7].

In late 1990, the Korean government considered how to reduce building energy consumption and minimize effects on the environment. Therefore, Green Standard for Energy and Environmental Design (G-SEED), known as the Korea version of Leadership in Energy and Environmental Design (LEED), came to be used during the building design phase. At the same time, a ZNE certification system and environmental impact assessment were utilized to reduce the carbon footprint of construction materials [8]. In addition, the government invested in R&D expenses to support construction, energy-saving, and eco-friendly buildings (housing complexes), so that related technologies could be commercialized in cities, which included Seoul and Jeonnam in South Korea [9].

The system must be recognized as essential to creating residential complexes in Seoul such as the ecological area rate. This method needs to mitigate the volume ratio through green building certification and the energy efficiency class certification system. Class 1 allows for up to 12% in volume and height limits, Class 2 is 8% relaxed, and if you obtain ZNE building certification, you may be permitted a floor area ratio of up to 15% max. The way the government directly supports the budget is by disseminating it through best practices, but it is also a burden for the government over the long term [10–12].

The Korean government has established and demonstrated an environmentally friendly system in various aspects, which include the distribution of building materials as R&D projects. The Ministry of Land, Infrastructure and Transport in Korea has extended an Environmental Product Declaration (EPD) to all construction materials, and these materials should be reflected in the entire eco-friendly architecture system, such as G-SEED, to supplement the system.

*1.1. Project Background*

As a practical project, Korean governmental ministries, such as the Ministry of Land, Infrastructure and Transport and the Ministry of Trade Industry and Energy, are promoting strategic projects to support cost-saving and low-carbon technologies in residential complexes and commercial buildings in the City of Suwon, South Korea. Suwon City is located in the mid-west portion of the Korean peninsula, has a population of approximately 1.2 million people, and covers an area of 47 square miles [13].

Suwon City has many government-affiliated institutions. According to the policy of the government to disperse the population in metropolitan areas, many government entities or their affiliates sell their land to the private sector and relocate. Most of the land sold to the private sector is bought by those with the intention of developing a large-scale residential complex. Although the sites, which are planned as large-scale residential complexes, are being constructed as eco-friendly complexes, in reality, most of the visible areas, such as the enlargement of landscaping areas, use artificial ground with a high volume ratio to recover development costs.

For these reasons, Suwon City plans to build a residential complex by combining apartment buildings and single-family houses. The lands are selected as a high-value residential complex site, using ecological planning techniques, such as low-carbon energy self-supporting apartments. The City of Suwon is the first case of a local government rolling out a budget and building an ecological village using the ZNE concept. Figure 1 presents the overall project locations, which include possible project sites (four dots) for this research project.

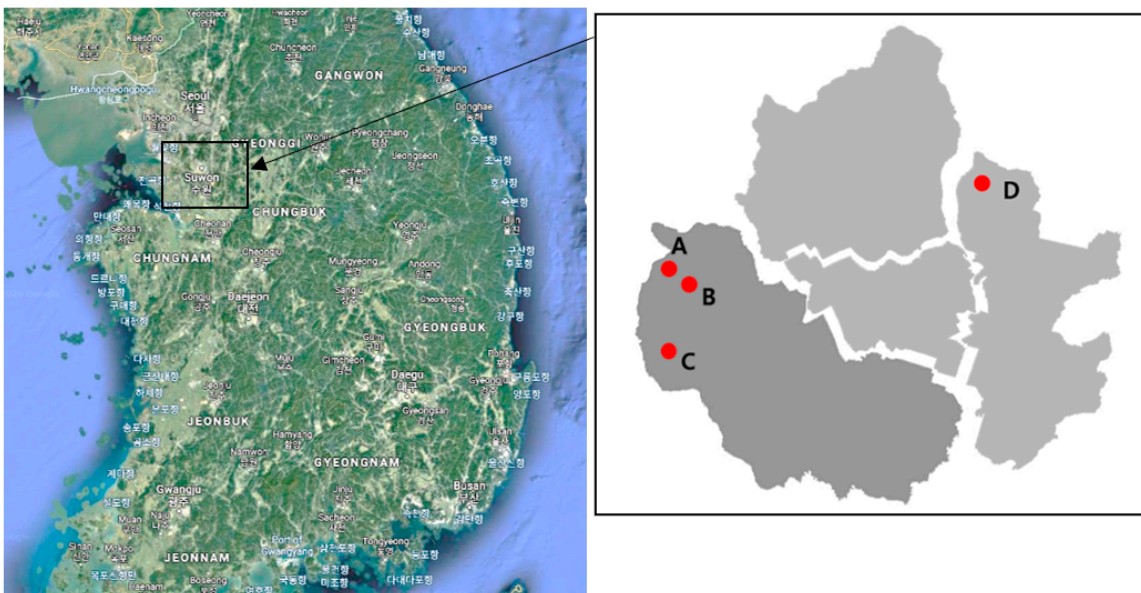

**Figure 1.** Overall Project Features, City of Suwon in South Korea (The four dots represent the areas to be selected for this research study).

These sites for the ZNE village project were selected among government-owned land to create a residential complex taking into account Suwon's unique identity. The Zero Net Energy Type Suwon Ecological Village Promotion Plan was created with this background. The plan is currently being applied to the development project that the public corporation promotes. The Korean government's role is crucial for the well-being of the construction-related technologies that fundamentally reduce energy use and reduce environmental impact.

*1.2. Research Objectives*

Buildings with eco-friendly building technology are built based for the causes of "climate crisis response", "carbon reduction", and "environmental protection", but they often end as one-time events because they are not economical. Therefore, Suwon City will be built considering economic feasibility while selecting performance targets and applied technologies so that eco-friendly building technology can be continuously expanded throughout the city. These efforts were reflected in the central government and other local governments as an exemplary attempt. Furthermore, it has developed from the material generation stage of the building as research that promotes carbon reduction during the building's entire life cycle.

This research study endeavors to re-examine the government's role in the universalization of energy and environmental technology through case studies where these technologies were applied to actual residential complexes. The objectives of this research study are: (1) to establish a land use plan for the project site in the City of Suwon, (2) to establish an external space plan, which includes the environmental aspects, and (3) to have alternative designs through a multi-criteria decision-making process.

**2. Previous Research Studies**

Over the years, extensive research studies on understanding of eco-friendly buildings, design techniques, building performance, optimization, and evaluation have been performed by various researchers. This research study identifies the gaps in previous studies between different aspects. The eco-village project in the City of Suwon, South Korea, is evaluated based on ecological and environmental aspects as well as cost analysis, and is the case study for establishing eco-villages in South Korea.

### 2.1. Concept of Community and Eco-Village Development Plans

Eco-village development has been encouraged because of its holistic concept of social and cultural resilience. Eco-village development was defined as a fully functional human-scale settlement with harmonious development, which is integrated into the environment and to be continued into the indefinite future [14,15]. This concept was expanded and redefined as a specific category of a society designed to regenerate social and natural environments and integrate cultural and economic dimensions in a long-term process [16,17]. As the eco-village concept has been formulated several times and in many different ways, implemented examples can be found in many places with great diversity worldwide [18,19].

Developing the local economy as a counter-force of economic globalization is a way to seek local economic autonomy, and this includes making economic decisions at the local level, developing locally owned business by preferably using local resources, employing local workers, satisfying the needs of local consumers, and supporting the local finance community [20]. The intention is to become more self-sufficient in production and consumption. Initiatives in this vein also include the relocalization of food production and consumption, and local complementary currencies is developed in order to keep wealth within the community [21–23].

Regionalization facilitates a more democratic approach to the economy, reduces unemployment, increases participation (and therefore integration), encourages solidarity, opens up new perspectives for developing countries, and finally, improves citizens' health in rich countries by encouraging sobriety and reducing stress [24].

Another dimension of relocalization, complementary to or as a consequence of economic localization, is bioregionalism. A bioregion is a region defined by natural boundaries considering geographic, climatic, hydrological, and ecological characteristics capable of supporting unique human and non-human living communities [25]. A bioregion has a high capacity for ecological self-sufficiency in terms of basic resources and is self-sustaining in terms of being in perfect harmony with the ecosystem and ensuring awareness of where resources come from and where waste goes [26]. Local agriculture should be protected and local renewable energy sources are encouraged to develop [25]. The potential benefits of embedding human activity within a bioregion are high energy efficiency, enhanced environmental sensitivity and accountability, and healthy human social relationships [20].

Both urban villages and eco-villages aim to realize the lifestyle imagined in a degrowth society by promoting certain spatial organization modes. By avoiding large concentrations of human beings, decentralized human-scale settlements can be better linked to their environment and reconcile humans with nature. Self-sufficiency can be achieved through farms within the village or in the nearby suburbs [27].

### 2.2. Eco-Friendly Technologies in Korea

The role of the government is essential in developing and applying eco-friendly technologies and commercializing them in Korea. Further, 16.7% of total land use in Korea is for urban areas and approximately 92% of Koreans live in these areas [9]. The housing type in Korea is likely a high-rise apartment rather than a single-family house or low-rise apartment. Even if people live private lives, apartments, where residents can utilize various community facilities such as cafeterias, gyms, markets, etc., in a complex, are preferred by most residents. Apartments are also recognized as a property investment, which can increase income through renting, so apartments are very important when considering the property and real-estate businesses [11].

The efforts to generalize eco-friendly technologies in Korea can be classified into three major categories: (1) a method to receive compensation for incentives, such as tax incentives and floor area ratio compensation, (2) a method for the government to seek out, create, and distribute green-related technology complexes, and (3) a method to support large-scale research and development (R&D) investment and link it with commercialization through demonstration [12].

### 2.3. Ecological Area Ratio and Energy Residential Towns

The ecological area ratio is an index that quantitatively induces the maintenance and improvement of natural circulation functions (evaporation, function, fine dust adsorption, stormwater permeability, and storage, soil, and habitat of animals and plants) to solve ecological problems in urban spaces [28]. It is an index designed to be used as a standard criterion for the quantitative evaluation of ecological values and each space type. It can be used as a pre-planning evaluation technique during the spatial planning stage. The ecological area ratio was implemented as a policy in Seoul in July 2004 and has been applied as a standard to assess environmental impact from 2015 [13,28]. The ecological area rate, which was created and published in 2016, is calculated by dividing the converted area by space type and the converted area by planting type by the total area. As the ecological area ratio results show the change in a city, this system, which can be applied and operated in units of buildings and apartments, was revised to be used in environmental impact assessments by the Ministry of Environment.

The Ministry of Land, Transport and Maritime Affairs in Korea has been pushing for mandatory zero-energy buildings for new buildings from 2025, reflecting the monitoring of the energy-saving effects of zero-energy housing [29]. "Zero energy" residential towns are divided into four stages in Korea, as follows [30].

- Eco-Town: Complex incorporating elements of energy consumption reduction and renewable energy.
- Zero Energy Town: The amount of the final energy consumed and the energy produced is the same; thus, no energy is needed from outside.
- Plus Energy Town: Surplus energy is generated throughout the complex.
- Energy Self-Sufficient Town: Energy supply is able to provide itself without being connected to the energy network.

### 2.4. Benefit to Cost (B/C) Analysis

Economic evaluations through cost-benefit analyses (BCA) typically estimate the total value of a particular project [16,31] and do not focus on specific attributes such as the value of improved accessibility or extended exposure to the project sites [32–34].

BCA is usually performed for all public projects, which includes Korean government research and development (R&D) projects, to ensure that there is no waste of the funds the government contributes [35,36]. In particular, a preliminary feasibility study ("Preliminary Feasibility Study for the Government R&D Projects and Fiscal Expenditures") must be submitted according to Article 38 of the National Finance Act [37].

In general, as a result of the preliminary feasibility study, if B/C ≥ 1, it means that there is economic feasibility, and if the analytic hierarchy process (AHP) ≥ 0.5, which means a comprehensive evaluation considers the policy feasibility, the project is deemed as being desirable. If the AHP score, which is calculated by reflecting economic feasibility, policy, regional balance, and technology, is 0.5 or higher, the preliminary feasibility study can be passed by the government [38].

The weight of each evaluation item is as follows unless otherwise noted. A reason must be provided for any alteration. If B/C analysis is performed according to the above background, in reality, Korea cannot secure more than 1 in B/C using eco-friendly technology because the cost of energy supplied to the public is too low. Excluding real estate costs, pure eco-friendly construction technology alone does not result in large B/C figures in the short term [39,40].

### 3. Materials and Methods: Land Use Plan for the Project Site in the City of Suwon, South Korea

Suwon City selected general residential areas and natural green areas in Gwonseon-gu after reviews of various sites. These lands were converted into residential areas for the creation of a zero-energy Suwon Eco-Village. The plans and proposed goals of Suwon City Eco-Village are shown in Table 1.

**Table 1.** Ecological Village Plans and Proposed Goals.

| | Aspects for Plans | Purpose |
|---|---|---|
| 1. | Eco-Environment: The organization for the integration of age and class, which are related to organic products, foods, etc. | • Growing organic crops<br>• Produce organic products<br>• Sell organic products and foods<br>• Consume organic products, produced, etc. |
| 2. | Establishment of target values. | • Green building certification standards applied<br>• Achieving 80% renewable energy<br>• Achieving 60% recycling rate<br>• Achieving 25% natural ground green area and 50% ecological area rate |
| 3. | Pedestrian-centered detached housing complex without cars on the ground. | • Creating a car-free ecological external space by creating an underground parking lot |

The eco-village in Suwon was established based on the organic agriculture's theme in three (3) main aspects to benefit the following: (1) economy and society, (2) ecological environment, and (3) energy. Among these aspects, the economy and society sectors were considered after the town was established, and the ecological environment and energy sectors were essential considerations for the town's development.

The purpose of the eco-environment sector is to minimize environmental damage, including terrain transformation and soil movement, and to create more and better ecological spaces. There are four (4) practical measures, which include rooftop greening, wall greening, permeable pavement, and natural greenery, to achieve these purposes. It also requires maintaining rainwater runoff up to at least 50%. Besides, another purpose of the eco-environmental sector is to build with harmless and healthy materials to create an ecological space. This ecological space ratio needs to be increased up to 50% of the land use and provides an environment where healthy food can be prepared in the village gardens.

The purpose of the energy sector is to prepare the basis for energy self-sufficiency, and it is a principle to meet the energy self-sufficiency rate up to 80% using renewable energy. To achieve this purpose, water-saving regulations must be implemented, including the use of water period, and the application of recycling waste with a recycling rate of at least 60%. In addition, solar power generators should be used and architectural structures should be created to minimize heat loss.

The eco-village in Suwon City designed all parking lots to be underground to have a car-free residential complex on the surface. This eco-village also considered a building-to-land ratio of less than 20% to limit the buildings of each site, thereby having more car-free ecological external space, as shown in Figure 2.

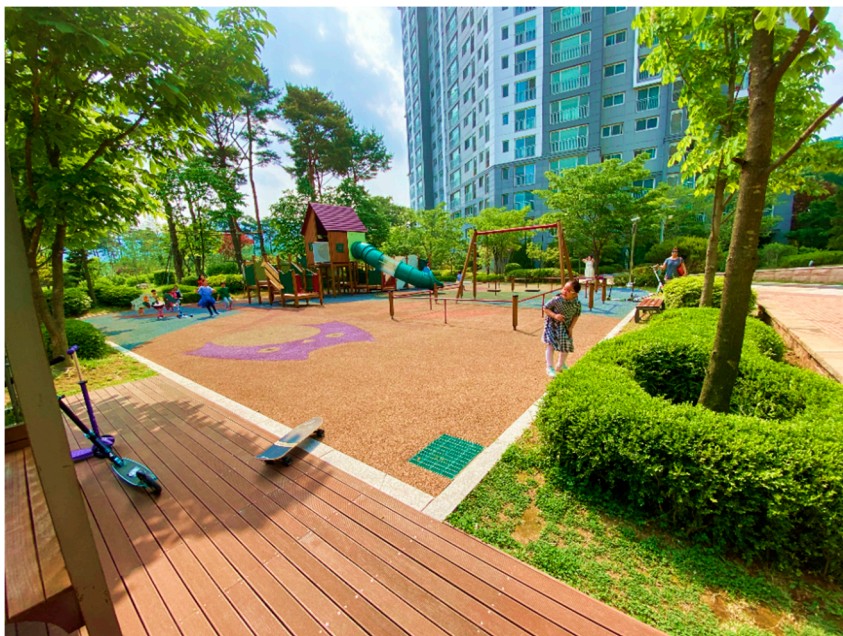

**Figure 2.** Car-free residential complex on the ground in Suwon.

## 4. Results: Ecological Planning and Environmental Aspects

To build and manage a better Suwon Eco-Village, domestic and overseas cases were reviewed. The project plan was set up through the Suwon Ecological Village Project, and operation and management were based on this review process. Figure 3 represents the Suwon Ecological Village Project process for this research study.

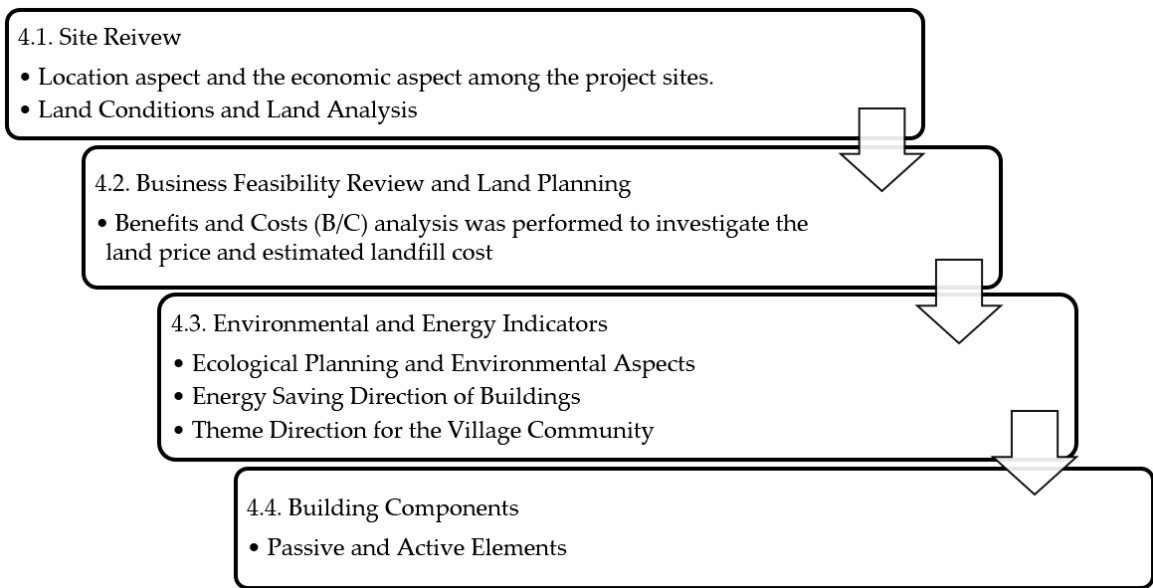

**Figure 3.** Project Development Process.

*4.1. Site Review*

Four (4) project site candidates were selected, which are in Suwon City. These sites were in a residential environment and had good accessibility to urban areas. In this research study, the four sites were defined as A, B, C, and D as shown in Figure 4.

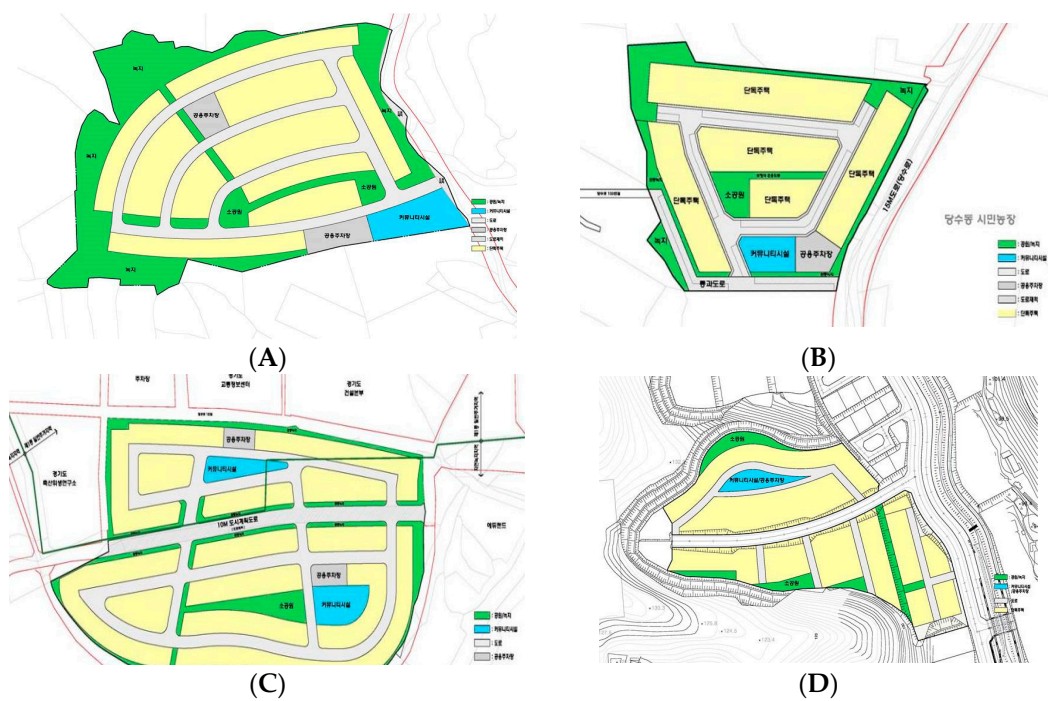

**Figure 4.** Alternative Sites in Suwon. (**A**) Kwonseon-Gu, Dangsoo-Dong 495-1; (**B**) Kwonseon-Gu, Dangsoo-Dong 437-6; (**C**) Kwonseon-Gu, Keumgok-Dong 791; (**D**) Yongtong-Gu, Iwi-Dong 1188.

These four (4) alternative sites were analyzed through location and economic aspects. First of all, in terms of the location condition, we surveyed the infrastructure utilization aspect, accessibility aspect, viewpoint, site elevation, and incentives from the government to select the sites. Table 2 shows the existing land conditions and land analysis of the selected sites. As a result of reviewing the above four sites, site C was considered the most suitable site. It also includes a large number of households when considering single houses and townhouses together.

*4.2. Business Feasibility Review and Land Planning*

In order to analyze the economic aspects of each candidate site, benefits and costs (B/C) analysis was performed to investigate the land price and estimated landfill cost. This is a widespread technique for evaluating a project or investment by comparing the economic benefits and economic costs of activity for government projects in South Korea. The economic assessment performed takes into consideration the case of a farm that outsources all field operations during the growing season and buys all production factors.

The largest business area is site A, and the area with the highest value area is site D. When calculating the unit price of these areas per square foot, the actual unit cost of site D is the highest as shown in Figures 5 and 6.

**Table 2.** Existing Land Conditions and Land Analysis among the Selected Sites.

| Alternate Sites | | A | B | C | D |
|---|---|---|---|---|---|
| Existing Land Condition | Local District | Green Area/Development Restricted Area | Green Area | Residential Area/Green Area | Residential Area |
| | Area | 55,015 | 13,156 | 52,626 | 26,311 |
| | Roadway Planning (Width) | 40′ | 50′ | 33′ | 33′ |
| | Cost (US $) to purchase land | 13.7 Billion | 9.6 Billion | 16.5 Billion | 32 Billion |
| Land Analysis | Surround Facilities | None | Moderate | Good | Good |
| | Accessibility | None | Moderate | Moderate | Moderate |
| | View | Adjacent to Chilbo Mountatin | None | Adjacent to Hosil Distrct. | Adjacent to Kwangkyo District. |
| | Location Direction | Southern | Southern | Southern | Southern |
| | Land Condition | Approx. 8.5% Slope | Approx. 3.1% Slope | Approx. 4.5% | Approx. 8.73% |
| | Number of Existing Dwells (Average house size) | 9 (Avg. house size: 570 SF) | 12 (Avg house size: 750 SF) | 200 (Avg house Size: 3300 SF) | 58 (Ave house size: 2850 SF) |
| | Remarks | • Need to secure infrastructure and improve public transportation accessibility.<br>• Concerns over prolonged business, such as high percentage of private land.<br>• The developed restriction zone needs to be lifted. | • Need to secure infrastructure and improve public transportation accessibility.<br>• High percentage of City land. | • Easy to expand the infrastructure and use of public transportation.<br>• Need to process the business after the public offices are relocated. | • Easy to expand the infrastructure and use of public transportation.<br>• Adjacent to Kwangkyo Mountain and nearby distribution of agricultural land. |

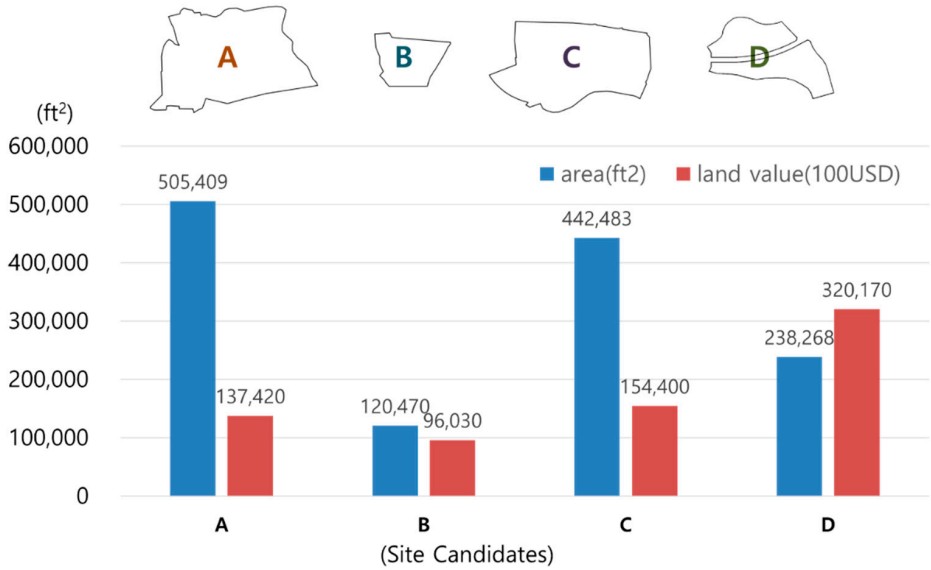

**Figure 5.** Land Price Comparisons among the Sites.

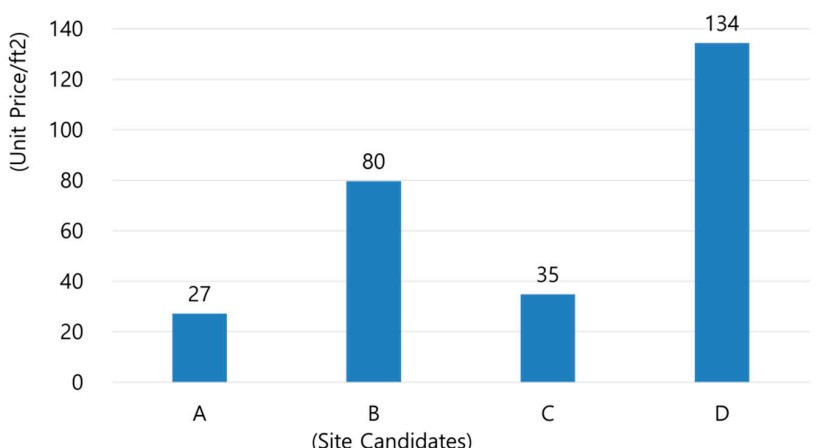

**Figure 6.** Unit Price/Square Foot among the Sites.

The estimated land share ratio for each site was calculated for each household under the assumption that the maximum number of households is built for each candidate site. The estimated land rent per household is shown in Figure 7.

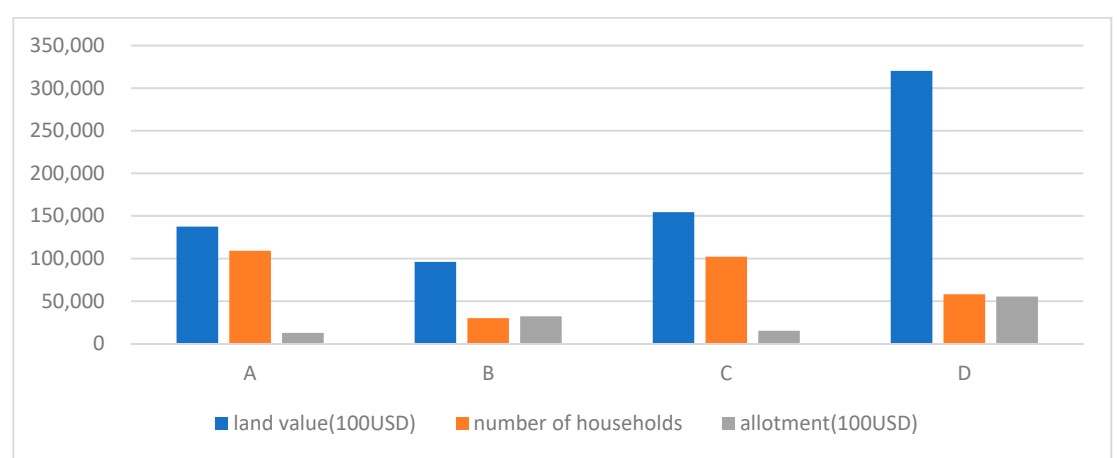

**Figure 7.** Estimated Land Rent per Estimated Household by Each Site.

B/C analysis was used to assess economic feasibility when the project was developed on each candidate site. In order to arrive at the infrastructure cost at each site, the development cost was calculated under the assumption that the infrastructure cost was the same as for nearby sites. In addition, sale prices were calculated based on comparable properties in the area. The energy-saving unit price of $6500/pyeong, (a unit of area in Korea, 1 pyeong = 35.5 $ft^2$), was applied to calculate the construction cost per household, and the price was based on a 33 pyeong nearby apartment.

Based on B/C analysis, site C showed a higher sales and distribution margin. However, site C had limited conditions to improve business feasibility, which was needed to construct townhouses rather than single houses. In site D, with all other infrastructure, the results of B/C analysis showed the best site conditions; however, the investment costs that individual residents had to pay were too high to develop the site.

Using B/C analysis for sites A through D, site C was selected as the most suitable site for the project by combining the location condition and the economic condition even though there were limitations. When the single houses were mixed with the apartments in site C, the B/C analysis results were higher than for other sites where only the single houses were developed. These two development directions were examined more closely. The B/C analysis results for alternative project sites are shown in Table 3.

**Table 3.** B/C Results for Alternative Project Sites.

| Alternative Sites | A | | B | | C | | D | |
|---|---|---|---|---|---|---|---|---|
| | Area, $m^2$ ($ft^2$) | Cost, $ | Area, $m^2$ ($ft^2$) | Cost, $ | Area, $m^2$ ($ft^2$) | Cost, $ | Area, $m^2$ ($ft^2$) | Cost, $ |
| Sale Price (B) | | 22,025 | | 7181 | | 26,927 | | 33,123 |
| Compensation + Infrastructure Costs (C) | 46,954 (505,401) | 31,452 | 11,192 (120,470) | 93,565 | 43,956 (473,138) | 48,791 | 22,136 (238,270) | 65,141 |
| B/C Feasibility | 1.0 | | 0.83 | | 1.0 | | 1.11 | |

### 4.3. G-SEED: Category 2–Energy and Environment

Various definitions of eco-friendly buildings or eco-villages in Korea were integrated into a quantitative certification system called G-SEED, which is similar to LEED, which is used in the U.S. and is applied to general residential buildings, apartment houses, commercial buildings, and school facilities. If buildings follow the certification system, various advantages are provided: (1) floor area ratio incentives, (2) financial support from the government, and (3) tax benefits.

Suwon Eco-Village borrows the concept of Class 1 green buildings, the highest performance of eco-friendly buildings certified by the state, but in the energy-saving sector, it starts at the 3-liter house, which is the level of realizable building energy. This 3-liter house is an energy-saving house with less than 3 liters per square meter (based on fossil fuel). This 3-liter house can save more than 80% compared to a 16-liter house. The 3-liter house building technology was applied with external wall insulation, triple-paned glass, high-precision windows, waste heat recovery devices, rooftop greening, and small solar and wind power generation systems.

In G-SEED, there are 7 main categories, which include (1) Use of Land and Transportation, (2) Energy and Environment, (3) Materials and Resources, (4) Water Efficiency, (5) Certification Maintenance, (6) Ecological Environment, and (7) Indoor Environmental Quality to be certified. Based on the checklist, this eco-village project focused on Category 2 (Energy and Environment). The results are evaluated in Table 4.

**Table 4.** Category 2- Energy and Environment from G-SEED.

| Category | Integrative Process | Credit | Residential Buildings | Commercial Buildings | School Facilities |
|---|---|---|---|---|---|
| 2. Energy and Environment | Ecological Planning and Environmental Aspects | Required | X | X | X |
| | Energy Saving for the Direction of Buildings | Required | X | X | X |
| | Theme Direction for the Village Community | Required | X | X | N/A |
| | Building Components | Required | X | X | X |

### 4.3.1. Ecological Planning and Environmental Aspects

The Ministry of Land, Infrastructure and Transport and the Ministry of the Environment recommend quantitative indicators in accordance with government green building certification standards to prevent any possible disputes related to the selection of indicators through national quantitative indicators. According to the G-SEED, the following is a summary of the proposals to attain Class 1 certification:

- Achieved 25% or more of natural ground green area.
- Earned a 50% ecological area ratio.
- Construction material sharmless to the environment and health.

In the case of the natural ground green area rates in Korea, it is difficult to exceed 20% in residential complexes to receive G-SEED Class 1 certification; however, the currently planned residential complex natural land ratio was 40.5%, which was approximately 20% higher than the standard.

### 4.3.2. Energy Saving for the Direction of Buildings

Through the basic design to minimize heat loss in winter and heat gain in summer, the building energy for a 3-liter house was reduced by 60% through the use of a passive system using natural energy and Class 1 energy efficiency. This passive system has an innovatively higher energy efficiency rating than the existing house's energy efficiency rating of 7 (17 L). When setting the energy efficiency level for a 3 -liter house, the heating cost and carbon emission reduction are calculated (see Table 5).

**Table 5.** Annual Heating Energy Savings Comparisons.

| | Annual Heating Energy Savings of 3 L Houses Compared to 17 L Houses: | Annual Heating Energy Savings of 3 L Houses Compared to 12 L Houses: |
|---|---|---|
| **Saving Cost** | $1250 <br> 105 m$^2$ × 14 L/m$^2$ × $1 × 85% of energy efficiency = $1250 | $803 <br> 105 m$^2$ d7 9 L/m$^2$ d7 $1 d7 85% of energy efficiency = $803 |
| **Saving CO$_2$** | 50,245 kg <br> 105 m$^2$ d7 14 L/m$^2$ d7 34.18 kg = 50,244.6 kg | 27,688 kg <br> 105 m$^2$ d7 9 L/m$^2$ d7 29.3 kg = 27,688 kg |

### 4.3.3. Theme Direction for the Village Community

To create an eco-village, village community themes such as "Organic village" and "Cultivate organic crops" were proposed to promote residents to produce and consume organic food simultaneously, suggesting a rural village within the city. With the above direction, the planning goals of Suwon Eco-Village are as follows:

- Housing Complex: Proposed directions for desirable future residential complexes, reflecting Suwon's past, present, and future, presenting a residential complex that reflects the values.
- Energy Zero: Suggestion of energy-efficient buildings and complex practical structure models; a model composition that can be introduced into new cities with high demand for new construction.
- Green Lifestyle: Presenting a green-life practice model, suggest a way to improve the value of life while coexisting with nature.
- Mobility: Building a transport system centered on pedestrians and public transportation, contributing to future energy savings by reducing transportation demands.

### 4.4. Building Components

Each building in Suwon Eco-Village has utilized a combination of passive and active elements to increase energy performance for the following:

- Passive elements:

  o South-facing location, high insulation, high-performance windows, waste heat recovery devices, external shading devices, heat bridge blocks.

- Active elements:

  o Water substation facilities, lighting fixtures, electrothermal equipment, heat source facilities, ventilation equipment, solar panels.

Passive buildings generally refer to buildings that do not need universal heating in the winter to maintain proper indoor temperatures, and supply the minimum amount of fresh air needed for daily life, using the minimum equipment, such as reusing natural heat and sunscreens for shading. The plan was carried out to cover additional energy consumption by energy generated through renewable sources, such as solar energy. Based on the combination of passive and active elements, the energy performance of each building was improved as shown in Figure 8.

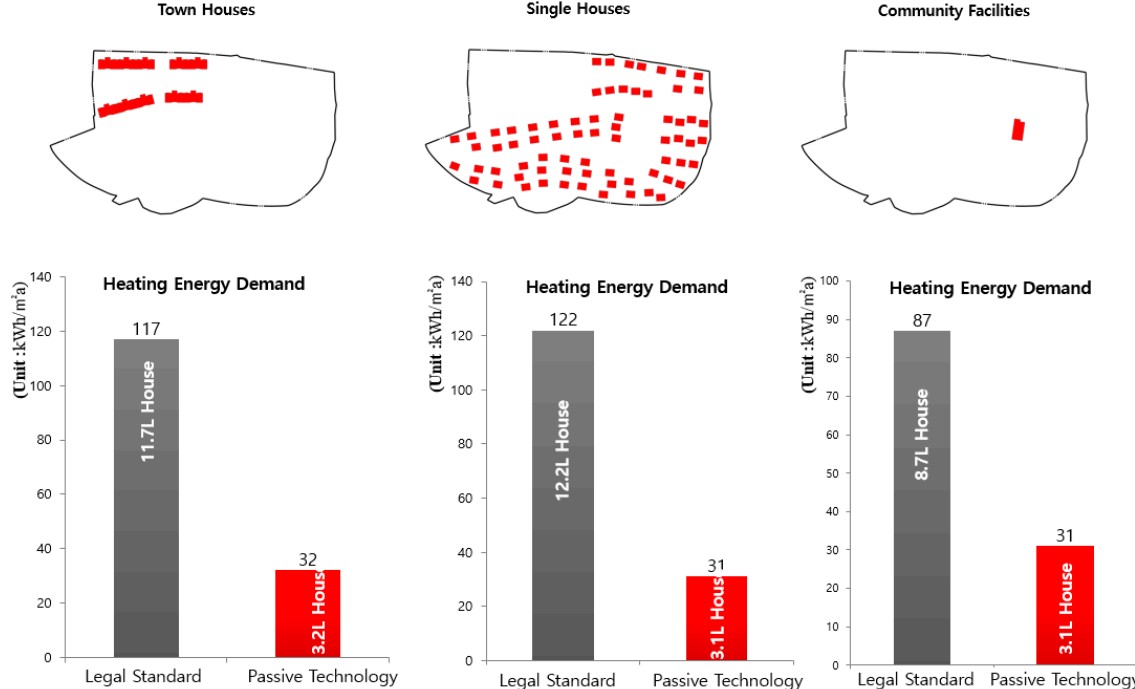

**Figure 8.** Improved Energy Performance Comparison.

If the buildings per parcel planned to reflect the current legal standards (based on the central part of Korea) are changed to passive performance standards through the application of passive elements, the annual heating energy (kWh/m$^2$a) requirements for houses improve by approximately 75% from 122 kWh/m$^2$a to 31 kWh/m$^2$a.

The energy simulation tool Passive House Planning Package (PHPP) was also used to confirm the annual heating energy required by houses. The results showed an improvement of 72 % from 117 kWh/m$^2$a to 32 kWh/m$^2$a. The community facilities were improved by approximately 64% from 87 kWh/m$^2$a to 31 kWh/m$^2$a. Annual heating energy composition refers to the energy performance of a building itself, and it can be seen that the building is comfortable without additional energy by using 3 liter of fuel consumption per year. The example of the PHPP for site C is shown in Figure 9.

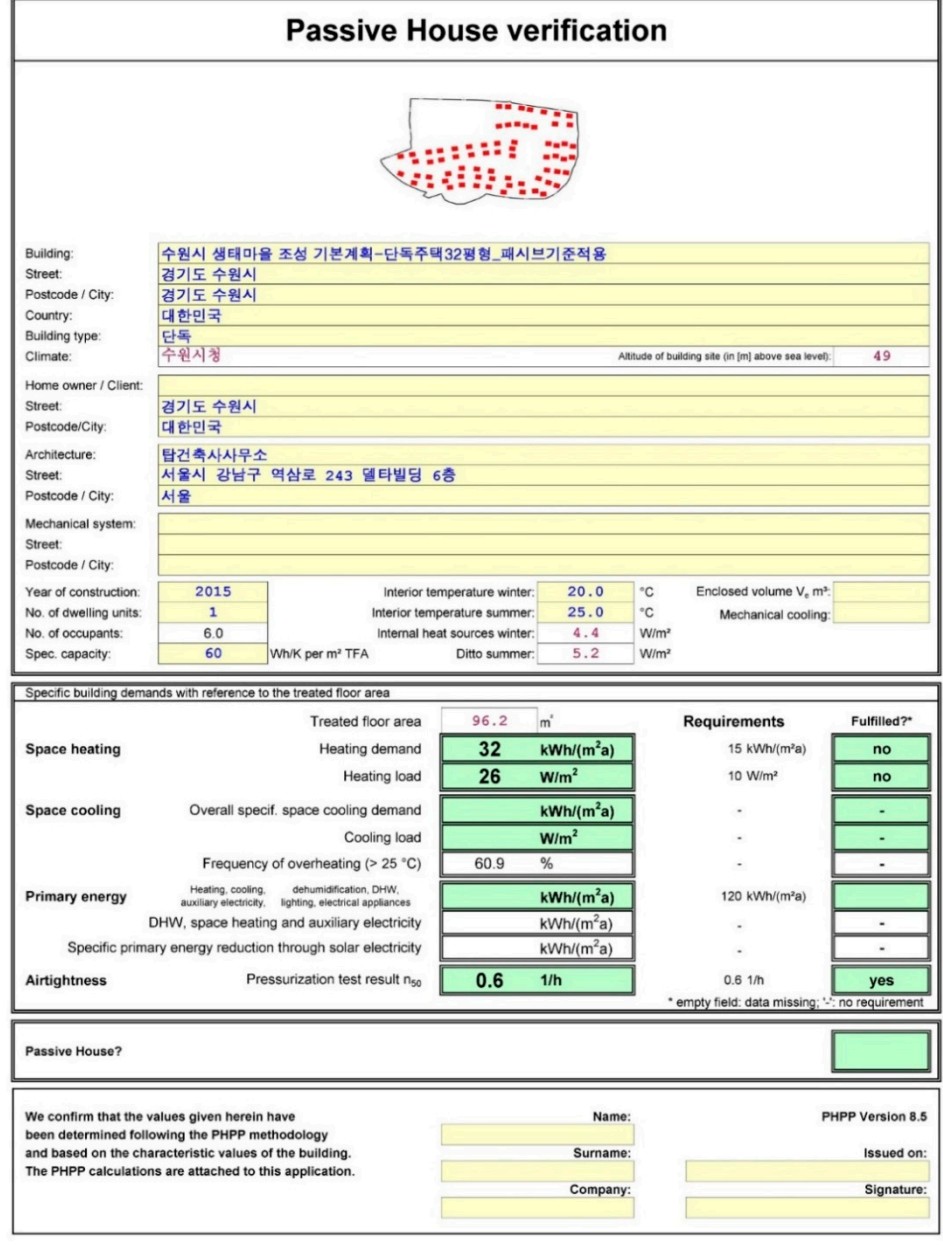

**Figure 9.** Example of the Passive House Planning Package (PHPP) for this Project.

## 5. Discussion and Conclusions

The attention given to the ZNE concept has increased during recent years in South Korea. Many countries have already established ZNE as their future building energy target. Among the

different strategies to save energy consumption, ZNEs have promising potential to significantly reduce energy use as well as to increase the overall market share of renewable energy. Existing technologies related to eco-friendly buildings were developed with policies and regulations to reduce energy loss in the construction field and minimize environmental destruction. However, eco-friendly technologies must be quickly distributed and commercialized. As people's income levels and awareness of the environment have increased, they prefer to be eco-friendly, so these factors have begun to affect housing trends.

The ecological variety of strategies used by Suwon City will ostensibly become a manual for an ideal apartment house, external space, and energy conservation, but an increase in construction costs will be inevitable. The purpose of this case study was to make a residential complex where newlyweds can move in and own an apartment in Suwon City. In order to preserve business feasibility at a cost 1.4 times more expensive than ordinary construction costs, the government should immediately find a way to provide support to this initiative.

The case study in Suwon City shows that the Korean local government has prepared a basic planning process to spread eco-friendly residential complexes and tried to introduce and realize eco-friendly construction standards proposed by the central government. The central government actively established a system to promote eco-friendly construction technologies in the market and encouraged people to use eco-friendly construction. This case study was used to build a complex that applies eco-friendly construction technology supported by the Korean government and developed by research institutes under the slogan of "Luxury Environmental City Suwon". The target buyers are middle-class Koreans who want to buy an apartment as a residence.

This study was an example of site selection and construction based on economic feasibility. As a part of this study procedure, B/C analysis was used to conduct an economic feasibility analysis in consideration of future distribution in Suwon City. Through B/C analysis for sites A through D, site C was selected as the most suitable site for the project by combining the location, land conditions, and economic conditions. Site C also meets the requirement based for G-SEED Category-2, which includes 7 main categories such as the Use of Land and Transportation, Energy and Environment, Materials and Resources, Water Efficiency, Certification Maintenance, Ecological Environment and, Indoor Environmental Quality to be certified.

The city's efforts to create eco-friendly residential complexes have shifted to other attempts to become the basis for construction technology based on government R&D. The level of technology and demonstration in this case study have been raised to the next level to consider the entire eco-village complex life cycle, such as production–construction–maintenance–disposal.

This study reflected national R&D and laid the foundation for a rental housing project created by eco-friendly technology called the "EZ House" in Nowon-gu, Seoul, as the second phase of the project. Recently, there has been national R&D that combines eco-friendly and minimized carbon footprints in the overall construction process, construction, operations, and maintenance in building materials beyond applying eco-friendly technology.

**Author Contributions:** S.-Y.M., D.J., and H.S.K. conducted the data and designed the research. S.-Y.M., J.-Y.L., and J.K. drafted and revised the manuscript based on the research results. All authors have read and agreed to the published version of the manuscript.

**Funding:** This work was supported by a grant [20AUDP-B146511-03] from the Architecture & Urban Development Research Program funded by the Ministry of Land, Infrastructure and Transport of the Korean government.

**Conflicts of Interest:** The authors declare no conflict of interest.

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
