# Peer review of "Importance of Government Roles for Market Expansion of Eco-Village Development Plan Establishment Research: Case Study in the City of Suwon, South Korea"

_sustainability, doi:10.3390/su122410293_

Round 1

Reviewer 1 Report

Review Comments:

The topic of developing eco-friendly housing is of general interest. I do, however, find the level of scientific contribution rather low in this paper. It uses a real case from South Korea, but presents it very matter-of-fact: this is what we did (basically a cost-benefit analysis of the choice of site between four candidates). References are sometimes odd, outdated or not relevant for this context. Some examples:   

In the introduction, building energy consumption in the US is first mentioned as 36 % (based on reference 4 (Koeppel & Urge-Vorsatz, 2007) and 6 (Kim et al, 2017)). It then says “nearly half”, rounding off the figure 40 % (taken from reference 18, which is the US Energy Information Administration, 2016). There might be different figures, but mixing sources and timespans give a confusing picture. It is also not immediately obvious to the reader how these figures relate to eco village construction in South Korea. Updated information about state-of-the art housing construction in South Korea, including statistics, would have been very informative.

Having read the text several times, I’m still not sure if ZNE and NZE are the same thing, and how they relate to the existing (?) “Zero Energy Building 49 certification system and Environmental Impact Assessment” mentioned in line 49-50.

In line 63-70, the principle of “ecological area rate” is described. In the text, the authors refer to “Class 1” and “Grade 2” buildings. It seems like an odd terminology within the same system. Then there is a third category (if the building gets a Zero Energy Building Classification). To a reader not familiar with the system, this description is more confusing than clarifying.

My recommendation is that this paper needs a substantial improvement to be published, including

  1. an updated list of relevant references
  2. a better introduction to the situation in South Korea, directed at international readers
  3. a more elaborated discussion about relevance and significance of this case study, both in a South Korean context, and from an international perspective.

Author Response

I appreciate your comments. I addressed all your comments to update the manuscript. Please find the attached comments and resolution form for your info. 

Thanks, 

Jonghoon Kim  

Reviewer 2 Report

I have reviewed the manuscript entitled “Importance of Government Roles for Market Expansion of Eco-Village Development Plan Establishment Research: Case Study in the city of Suwon, South Korea.”  I feel that the paper offers interesting and timely topic. However, in its current form, it is insufficient in terms of addressing new theoretical arguments, explaining generic relevance, containing sufficient contributions to the new body of knowledge from the international perspective, and discussing implications for sustainable urban policies beyond the local case. Accordingly, the research design must be improved.

  • The introduction could be re-structured to highlight the contribution and the motivation of the paper.
  • The authors should add more literature review on theoretical and methodological approaches to examine the role of the government roles for expansion of eco-village development plan.
  • The authors should discuss the gaps in the literature they found.
  • Research design must be based on research objective. Figure 3 needs more explanation in the text. B/C analysis can be used for business feasibility for site selection, however, in an attempt to achieve the objective of this academic paper, rather the authors might find that you would need to do considerably more research to find out the appropriate methodology.
  • Regarding “4.3. Environmental and Energy Indicators” how the authors select the indicators? What was the literature background?
  • Sentences remain unclear and the methodology and discussion of the paper is a bit difficult to follow for those who are not familiar with the City of Suwon in South Korea. A better series of maps and figures would help.
  • In the results part, the authors should discuss how the results can be interpreted in perspective of previous studies and need to specify those observations in the context of Suwon environment. Regarding the conclusions a deeper “generalization” of the results could also be implemented. The authors could consider giving more importance and space to describe how the research results could help other experiences and what this study entails for the field of research.
  • In the conclusion section, I would like to see the paper to point out a number of limitations of the research as well as some future research avenues.

Author Response

(The authors gave the same response as above.)

Round 2

Reviewer 2 Report

You have addressed the issues raised to  some  extend. So I won't oppose the publication of this paper.